

# A retrospective study: analysis of the relationship between lactate dehydrogenase and castration-resistant prostate cancer based on restricted cubic spline model

Ruiying Qiu[1,*], Ke Bu[1,*], Hengqing An[2] and Ning Tao[1]

[1] Department Public Health, Xinjiang Medical University, Urumqi, Xinjiang, China
[2] The First Affiliated Hospital, Xinjiang Medical University, Urumqi, Xinjiang, China
[*] These authors contributed equally to this work.

## ABSTRACT

**Background**. Different prostate cancer patients take different amounts of time to progress to castration-resistant prostate cancer (CRPC), and this difference in time determines the patient's ultimate survival time. If the time to progression to CRPC can be estimated for each patient, the treatment can be better individualized.

**Objective**. Castration-resistant prostate cancer is a challenge in attacking prostate cancer, the aim of the paper is to analyze the correlation between lactate dehydrogenase (LDH) and CRPC occurrence based on the restricted cubic spline model, and to provide a theoretical basis for LDH as a prognostic biomarker for prostate cancer patients.

**Methods**. We retrospectively analyzed clinical and follow-up data of patients diagnosed with prostate cancer and treated with Androgen Deprivation Therapy (ADT) in our hospital from October 2019 to August 2022. Investigate the correlation between LDH and CRPC by COX regression, restricted cubic spline model and survival analysis.

**Results**. The initial tPSA concentration, prostate volume, LDH and alkaline phosphatase levels in patients with prostate cancer with rapid progression are higher than those in patients with prostate cancer with slow progression. Multivariate COX regression showed that initial tPSA level and LDH level are independent risk factors for prostate cancer. Restricted cubic spline model further showed that LDH level is linearly correlated with the risk of CRPC in prostate cancer patients (total $P < 0.05$, nonlinear $P > 0.05$).

**Conclusion**. LDH was associated with the prognosis of prostate cancer and had a dose-response relationship with the risk of CRPC in prostate caner patients.

# INTRODUCTION

Prostate cancer is a major public health problem worldwide, according to the American Cancer Society, prostate cancer will have the highest incidence and the second highest mortality rate among male cancers in 2022, the mortality rate will be second only to

Corresponding authors
Hengqing An, 9269735@qq.com
Ning Tao, 18167968803@163.com

lung cancer (*Siegel et al., 2022*). At present, the difficulty in the prevention and treatment of prostate cancer is the onset is insidious, patients are usually first diagnosed when they are in a middle or advanced stage, the middle and advanced stage prostate cancer usually progresses rapidly to castration-resistant prostate cancer (CRPC) after androgen deprivation therapy (ADT) (*Cheng et al., 2022*), and the average survival time of CRPC patients is only 2~3 years (*Chang et al., 2014*; *Ritch & Cookson, 2018*). Clinical studies have shown significant differences in the time to progression to CRPC after ADT in patients with different degrees of prostate cancer (*Lin et al., 2019*; *Nair et al., 2021*), this suggests that different prostate cancer patients have different sensitivities to ADT, and studying the difference in the time of progression to CRPC in different prostate cancer patients can better predict the benefit degree and prognosis of prostate cancer patients with ADT, which has certain significance for the clinical diagnosis and treatment of prostate cancer patients.

At present, the factors influencing the length of time it takes prostate cancer to progress to CRPC are not conclusive, factors such as age, Prostate Specific Antigen (PSA) level, Gleason score, T stage, and alkaline phosphatase level have been associated with the development of CRPC (*Koo et al., 2015*; *Tan et al., 2021*). With the deepening of the research on tumor metabolic pathways, it is found that glycolysis is the main energy supply mode of tumor, and it is closely related to the occurrence and development of prostate cancer (*Abdel-Wahab, Mahmoud & Al-Harizy, 2019*). Lactate dehydrogenase (LDH) is the key enzyme of glycolysis pathway (*Xian et al., 2015*), and may be a prognostic biomarker in the clinical treatment of prostate cancer (*Mori et al., 2019*). Most studies have shown that LDH has a diagnostic value for prostate cancer, and no study has pointed out that LDH has a correlation with the progression-free time of prostate cancer; however, the length of time for prostate cancer to progress to CRPC is an important indicator for determining the effectiveness of treatment and prognosis of prostate cancer patients.studies have shown that expression level of LDH is related to the occurrence and prognosis of cancer. This study explores the correlation between LDH and CRPC occurrence, and provides a theoretical basis for LDH as a prognostic biomarker for prostate cancer patients.

## MATERIALS & METHODS

### Subjects

The study took place in the urology surgery in first affiliated hospital, Xinjiang Medical University. We selected the prostate cancer patients meet the following criteria (1) The patients were diagnosed with prostate cancer by pathological biopsy and progressed to CRPC stage; (2) The patients were treated with ADT in our hospital after diagnosis of prostate cancer; (3) The patients were first-time prostatectomy at the time of diagnosis of prostate cancer in our hospital; (4) The patients had complete clinical and follow-up data, voluntarily participated in the study, and signed the informed consent form (Ethics Approval No. 20180223-166). Exclusion criteria: (1)The combination of other malignant tumors; (2) Incomplete clinical or follow-up data; (3) The patients were not eligible for or did not receive ADT treatment; (4) Patients underwent prostate cancer resection after diagnosis. A total of 147 patients who met the criteria were finally included in this study.

<br>

## Therapeutic regimen

In this study, there were three ADT regimens for prostate cancer patients: (1) Goserrelin 3.6 mg/Leuprelin 3.6 mg subcutaneously once a month; (2) Goserrelin 3.6 mg once a month subcutaneously and Abiraterone 1,000 mg once a day orally; (3) Subcutaneously 3.6 mg Goserillin once a month and Bicalutamide 50 mg once a day orally. We primarily used the patient's T-stage, insurance, and financial status to make treatment plan choices.

## Trial design

We collected basic data of the patients, we collected basic data, including nation, age, BMI, history of diabetes, hypertension, smoking, family tumor history, Gleason score, tumor stage, and therapeutic regimen. Peripheral venous blood of all patients was collected on an empty stomach from the morning of diagnosis after admission to detect PSA, testosterone, LDH, alkaline phosphatase and other indicators. We divided the patients into the rapid progression group and the slow progression group according to the median survival time of the subjects, preliminarily determined the factors that may be related to the progression of CRPC by comparing the differences of relevant variables between the two groups. After determining that there was no difference in survival time and LDH content among patients with different therapeutic regimen, we used COX regression to further clarify the influencing factors of CRPC occurrence time. Then we constructed a restricted cubic spline model of lactate dehydrogenase and CRPC based on the results of COX regression, to explore the dose–response relationship between lactate dehydrogenase content and CRPC.

## Statistical analysis

Using the Kolmogorov Smirnov method to test the normality of continuous variables, non-normally distributed continuous variables are represented by median and quartile, the rank-sum test was used to analyze the difference of measurement data that did not conform to normal distribution, the rate represents the categorical variable, chi-square tests were used to compare categorical variables between groups, using COX regression analyze the correlation between all factors and CRPC occurrence. Based on the results of COX regression, the dose–response relationship between lactate dehydrogenase and CRPC occurrence risk after adjusting the initial tPSA group was analyzed by restricted cubic spline analysis. $P \leq 0.05$ was considered as statistically significant difference. All statistical analysis was carried out by R4.1.1 software (*R Core Team, 2021*).

## RESULTS

### Study population characteristics

The median survival time of all patients in this study is 18.830 (17.231~20.429) months. Divided the patients into a rapidly progressing group ($n = 87$) and a slowly progressing group ($n = 60$) by median survival time of 19 months. The demographic, health status, lifestyle, and prostate cancer disease characteristics of the 147 men enrolled in this study are summarized in Table 1, there are significant differences in initial tPSA, prostate volume, lactate dehydrogenase and alkaline phosphatase between the two groups ($P < 0.05$).

**Table 1  Basic information of prostate cancer patients.**

| Factors | N | Rapidly progressing group | | Slowly progressing group | | $\chi^2/Z$ | P |
|---|---|---|---|---|---|---|---|
| | | n | % | n | % | | |
| Age (year) | | | | | | | |
| ≤65 | 32 | 21 | 65.63 | 11 | 34.38 | | |
| 66∼80 | 91 | 49 | 53.85 | 42 | 46.15 | 2.917 | 0.226 |
| >80 | 24 | 17 | 70.83 | 7 | 29.17 | | |
| Nation | | | | | | | |
| Han | 81 | 42 | 51.85 | 39 | 48.15 | 4.015 | 0.063 |
| Minority nationality | 66 | 45 | 68.18 | 21 | 31.82 | | |
| Initial tPSA (ng/mL) | | | | | | | |
| ≤200 | 65 | 29 | 44.62 | 36 | 55.38 | 10.238 | 0.001 |
| >200 | 82 | 58 | 70.73 | 24 | 29.27 | | |
| T stage | | | | | | | |
| T2 | 36 | 21 | 58.33 | 15 | 41.67 | | |
| T3 | 36 | 17 | 47.22 | 19 | 52.78 | 3.317 | 0.190 |
| T4 | 73 | 49 | 65.33 | 26 | 34.67 | | |
| Gleason scores | | | | | | | |
| ≤7 | 18 | 10 | 55.56 | 8 | 44.44) | | |
| 8 | 63 | 43 | 68.52 | 20 | 31.75 | 6.010 | 0.111 |
| 9 | 46 | 21 | 45.65 | 25 | 54.35 | | |
| 10 | 20 | 13 | 65.00 | 7 | 35.00 | | |
| Prostate volume (mL) | | | | | | | |
| ≤40 | 46 | 28 | 60.87 | 18 | 39.13 | | |
| 40∼60 | 50 | 23 | 46.00 | 27 | 54.00 | 6.056 | 0.048 |
| >60 | 50 | 35 | 70.00 | 15 | 30.00 | | |
| ADT regimen | | | | | | | |
| Goserelin/Leuprorelin | 50 | 29 | 58.00 | 21 | 42.00 | | |
| Goserelin+Abiraterone | 50 | 33 | 66.00 | 17 | 34.00 | 1.689 | 0.430 |
| Goserelin+Bicalutamide | 47 | 25 | 59.18 | 22 | 40.82 | | |
| LDH (U/L)[a] | | 193.10 | 165.10∼278.00 | 182.60 | 160.73∼207.25 | 2.288 | 0.022 |
| Alkaline phosphatase(U/L)[a] | | 207.50 | 107.20∼396.60 | 111.05 | 85.55∼209.73 | 2.842 | 0.004 |
| Testosterone (mol/L) [a] | | 12.48 | 9.24∼16.3 | 111.05 | 8.60∼15.40 | 1.143 | 0.253 |

Notes.
[a]Using median and IQR to describe.

## LDH in patients with different ADT regimens

There is no significant difference in the concentration of LDH and the time to progression to CRPC among the three treatments by one-way analysis of variance ($P > 0.05$), specific data are given in Table 2.

## Factors influencing the progression of prostate cancer to CRPC

Univariate COX regression shows that initial tPSA and lactate dehydrogenase are correlated with the risk of CRPC($P < 0.05$). Included the significant variables in univariate COX regression into multivariate COX regression, the result shows that the initial tPSA and

**Table 2** Analysis of lactate dehydrogenase and time of progression in patients with different ADT regimens.

| ADT regimen | LDH (U/L) | Z | P | Progress time (month) | Z | P |
|---|---|---|---|---|---|---|
| Goserelin /Leuprorelin | 184.40(163.58∼246.10) | | | 15.00(12.00∼24.25) | | |
| Goserelin+ Abiraterone | 194.55(163.33∼232.50) | 0.856 | 0.652 | 15.00(10.00∼20.25) | 5.001 | 0.082 |
| Goserelin+ Bicalutamide | 192.00(163.00∼251.00) | | | 18.00(13.00∼25.00) | | |

**Table 3** Univariate and multivariate COX risk regressions for progression of CRPC in prostate cancer patients.

| Factors | Univariate COX regression | | | Multivariate COX regression | | |
|---|---|---|---|---|---|---|
| | HR | 95%CI | P | HR | 95%CI | P |
| Initial tPSA (≤200 ng/mL) | 1.548 | (1.112∼2.156) | 0.010 | 1.470 | 1.050∼2.058 | 0.025 |
| Prostatic volume (≤40 mL) | | | | | | |
| 40∼60 mL | 0.810 | 0.542∼1.211 | 0.305 | | | |
| >60 mL | 1.198 | 0.799∼1.796 | 0.383 | | | |
| LDH (U/L) | 1.001 | 1.000∼1.002 | 0.003 | 1.001 | 1.000∼1.002 | 0.013 |
| Alkaline phosphatase(U/L) | 1.000 | 1.000∼1.001 | 0.281 | | | |

LDH are the factors influencing the onset time of CRPC($P < 0.05$), the specific results are shown in Table 3.

## Dose–response relationship between LDH and risk of CRPC occurrence

The results of the restricted cubic spline model showed a linear correlation between LDH and CRPC (linear trend $P = 0.004$, Nonlinear $P = 0.406$), when LDH concentration>191.5 U/L, the risk of CRPC in prostate cancer patients increased significantly with each 1 U/L increase in LDH, Fig. 1.

## Survival analysis of CRPC occurrence in patients with different LDH levels

Based on the results of the restricted cubic spline curve, when the LDH concentration was greater than 191.5 U/L, the patients' risk of developing CRPC increased with the increase of LDH, and divided the patients into two groups with a critical value of LDH concentration of 191.5 U/L. Comparing the average survival time between the two groups, it is found that the average survival time of patients with LDH ≤191.50U/L (20.487(17.926∼23.048) months) was greater than LDH > 191.50U/L patients (16.975(15.245∼18.668) months), the difference was statistically significant ($P < 0.05$), Fig. 2.

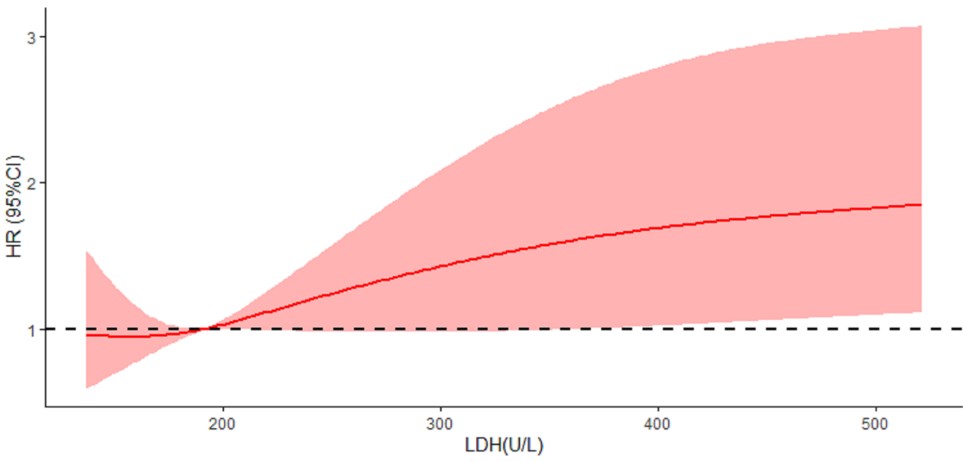

**Figure 1** Univariate and multivariate COX risk regressions for progression of CRPC in prostate cancer patients.

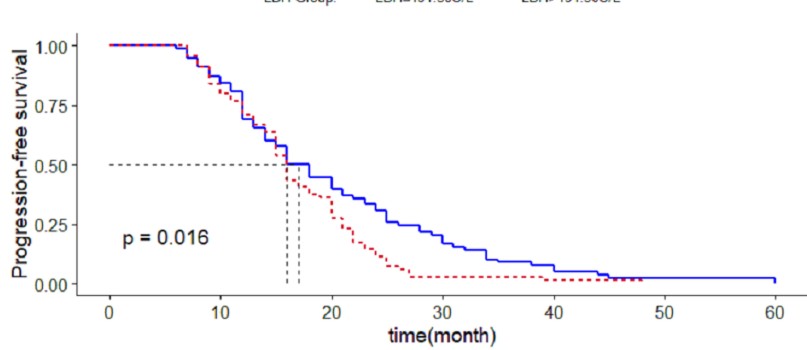

**Figure 2** Survival analysis of CRPC occurrence in patients with different lactate dehydrogenase groups.

## DISCUSSION

CRPC is a difficult problem in the prevention and treatment of prostate cancer. Currently, there is no recognized effective method for the prevention and treatment of CRPC in clinical practice. Therefore, it is of great significance to determine the risk of developing CRPC in prostate cancer patients as soon as possible, as well as the time of benefit in ADT, for guiding individual treatment and improving the prognosis of patients (*Turco et al., 2022*), therefore, the search for markers related to the occurrence of CRPC is the key to the follow-up treatment and prognosis of prostate cancer patients. Previous studies have shown that factors such as initial PSA level, alkaline phosphatase, prostate volume, and Gleason score can affect the time it takes for prostate cancer to progress to CRPC (*Nayyar, Sharma & Gupta, 2010*; *Benaim, Pace & Roehrborn, 2002*; *Ross et al., 2008*). In our study, only differences in prostate volume, initial PSA, alkaline phosphatase and LDH were found in prostate cancer patients with different progression rates, and further multifactor COX

regression confirmed that initial tPSA and alkaline phosphatase were associated with the risk of CRPC, higher initial tPSA and LDH were risk factors for CRPC. The results of this study show that LDH of patients at the time of diagnosis is closely related to the time of progression to CRPC of prostate cancer patients, and the higher the LDH level is, the higher the risk of CRPC occurrence is, and the two are linearly correlated. This indicates that the LDH level can predict the risk of CRPC in prostate cancer patients.

LDH is a key enzyme in the last step of the glycolysis pathway and is closely related to the glycolysis level of the tumor (*Jurisic, Radenkovic & Konjevic, 2015*), glycolysis is a way of normal cells under the condition of hypoxia, The study by German scientists, Otto Warburg found that unlike most normal tissue, tumor cells even in the case of oxygen content is normal, glucose intake and the accumulation of lactate will gradually rise, using glycolysis as the main source of energy metabolism, get higher sugar decomposition ability, make glucose to lactate to produce ATP, this phenomenon is called the Warburg effect (*Koppenol, Bounds & Dang, 2011*). This indicates that glycolysis is the main energy supply mode of tumor and plays a crucial role in the occurrence and development of tumor. The Warburg effect represents the transformation of glucose utilization by tumor cells from oxidative phosphorylation to glycolysis, which is considered as one of the most basic metabolic changes in the process of malignant transformation, and is also considered as a marker of malignant tumors (*Fe et al., 2018*). The Warburg effect has been confirmed in various malignant tumors such as liver cancer, stomach cancer and breast cancer. Such changes in energy metabolism are regulated by many complex factors, including the pressure of tumor microenvironment and gene changes. Due to the specificity of glycolysis in malignant tumors, some experts consider the development of anti-glycolysis drugs combined with anticancer drugs to improve the therapeutic effect (*Tan et al., 2021*).

Since the main function of the prostate is to secrete citrate, which serves as a source of ATP for sperm (*Ronquist et al., 2013*), normal prostate epithelial cells are rich in zinc ions, which can inhibit the normal aerobic oxidation pathway in cells (*Gumulec et al., 2011*). This results in the glycolysis being higher in normal prostate epithelial cells than in other normal cells. The difference of glycolysis level between prostate cancer cells especially in early tumor cells and normal prostate cells is little, but this characteristic does not affect the fact that glycolysis is still the main energy supply mode of prostate cancer, and the change of glycolysis level in cells still affects the occurrence and development of prostate cancer. LDH is a key enzyme in glycolysis pathway, its expression level is also correlated with the occurrence and development of prostate cancer. Some studies have found that the overexpression of LDH could predict the decrease of the overall survival of prostate cancer patients (*Mori et al., 2019*), moreover, LDH is related to the stage and grade of tumor, and has the value of assisting in the diagnosis of prostate cancer (*Yamada et al., 2011*). This indicates that LDH level not only has the potential to predict the effect of ADT in prostate cancer patients, but also may play an important role in the pathogenesis of CRPC. At present, most studies use COX regression model to analyze risk factors, but the COX regression model usually converts the continuous variables in the independent variables into classified variables for analysis, while the restricted cubic spline could analyses the dose–response relationship between continuous variables and dependent variables, the

influence of small changes in independent variables on the value of dependent variable HR can intuitively presented in the form of continuity curve (*Desquilbet & Mariotti, 2010*; *Liu et al., 2022*), we believe that the change of LDH within a certain range does not affect the progression-free survival of prostate cancer patients, and when LDH reaches a certain threshold, the risk of patients' progression to CRPC increases with the increase of LDH, so we chose the restricted cubic spline modeling to find this threshold of LDH, and the results of this study by restricted cubic spline modeling shows that when the concentration of LDH >191.5 U/L there is a linear correlation between LDH and the progression of prostate cancer, the higher the LDH level of the patient's prostate cancer, the higher the risk of CRPC and the faster the progression to CRPC. . The search for additional mechanisms of prostate cancer progression through LDH may also be considered in mechanistic studies of prostate cancer.

## CONCLUSIONS

In conclusion, LDH has a certain prognostic value for prostate cancer. It can be used to evaluate the risk of patients progressing to CRPC and the degree of benefit from ADT through its combination with PSA level in clinical practice, which is helpful for clinicians to select individualized treatment methods for prostate cancer patients. In the future, we can increase the sample size to study the correlation between LDH and CRPC, and study the mechanism of LDH in the occurrence and development of CRPC.

## ACKNOWLEDGEMENTS

The authors thank all patients and investigators.

### Funding

This work was supported by the Natural Science Foundation of Xinjiang Uygur Autonomous Region (No. 2022D01D39), and the Tianshan Elite Youth Program of Xinjiang Uygur Autonomous Region(No. 2022TSYCCX0026). The funders had no role in study design, data collection and analysis, decision to publish, or preparation of the manuscript.

### Grant Disclosures

The following grant information was disclosed by the authors:
Natural Science Foundation of Xinjiang Uygur Autonomous Region:  2022D01D39.
Tianshan Elite Youth Program of Xinjiang Uygur Autonomous Region: 2022TSYCCX0026.

### Competing Interests

The authors declare there are no competing interests.

## Author Contributions

- Ruiying Qiu conceived and designed the experiments, performed the experiments, analyzed the data, prepared figures and/or tables, authored or reviewed drafts of the article, and approved the final draft.
- Ke Bu conceived and designed the experiments, performed the experiments, analyzed the data, prepared figures and/or tables, and approved the final draft.
- Hengqing An analyzed the data, authored or reviewed drafts of the article, and approved the final draft.
- Ning Tao performed the experiments, prepared figures and/or tables, authored or reviewed drafts of the article, and approved the final draft.

## Human Ethics

The following information was supplied relating to ethical approvals (i.e., approving body and any reference numbers):

Xinjiang Medical University

## Data Availability

The raw measurements are available in the Supplementary File.

## Supplemental Information

Supplemental information for this article can be found online at http://dx.doi.org/10.7717/peerj.16158#supplemental-information.

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
