# Peer review of "A retrospective study: analysis of the relationship between lactate dehydrogenase and castration-resistant prostate cancer based on restricted cubic spline model"

_PeerJ, doi:10.7717/peerj.16158_

## Round 0.1 · original submission · Major Revisions

We have received three reviews requiring revisions. Reviewer #2 asked for additional experimental validation. Since your manuscript is a Review, you do not need to do experiments, just add proper references and discussion. I believe the remarks are constructive and you will update the work accordingly.

Reviewer 1 ·

Basic reporting

This study is a well-written manuscript and easy to follow.

Experimental design

Many cancers have been found to have elevated LDH levels, and their prognostic value has been shown to be of prognostic value in various solid and hematologic malignancies. In Prostate cancer, there have been also several studies investigating the relationship between LDH and oncologic outcomes.
This study aimed to investigate the relationship between LDH and CRPC using the restricted cubic spline model.
This study is a well-written manuscript and study design is clear.

Validity of the findings

The results were clearly presented and could be followed easily. Few comments.
-I think it’s better to add the data of ADT regimens in Table1.
-Are patients with M0CRPC included?
-It’s better to include hazard ratio, not just p-values.

Additional comments

The introduction is well-written and easy to follow.
There have been several studies investigating the relationship between LDH and oncologic outcomes. I think it would be better to clarify the novelty of this study.

Reviewer 2 ·

Basic reporting

The article needs proper grammar editing with appropriate punctuation. The authors wrote the paper for a focused group. Also, I would expect a little detailing of techniques used to quantify the data. I would like to read more about why the authors chose the cubic spline model compared to other statical models available.
The paper lacks substantial biochemical assays pointing to using LDH as a potential marker for CRPC. The result session should give a little more information about the data accumulated.

Experimental design

Given the importance of working on patient samples, the authors didn't use the precious opportunity to solidify their findings using biochemical assays. The work lacks supporting data for their discovery. This work provides evidence regarding LDH activity in prostate cancer cells but fails to bridge the gap in the focused research field. It is hard to correlate the methods described to the actual findings reported in the results section.

Validity of the findings

The work mentioned here has significant value to public health but needs more thorough follow-up experiments to support the findings.

Additional comments

The background studies pointed out by the authors show the recurrence of specific pathologies related to prostrate cancer, especially CRPC. The author's finding is exciting and shows how relevant it is to use LDH as a biomarker for CRPC. As it is essential to show why the study is conducted, it's also important to validate the findings shown by the authors. The results shown here need more validation. Until then, the results can be only stated as "potential findings."

Reviewer 3 ·

Basic reporting

The manuscript is clearly written in professional. While the introduction needs more detail. I suggest that explain the difference between regimens of the ADTs and add more references on the survival information about the ADTs.

Experimental design

The Trial design section needs more detail. I suggest that you improve the description at line 90 to 93 to provide more justification and explanation on the rapid progression group and the slow progression group separation.

Validity of the findings

Table 1 the proportion of the Nation rows need to be updated.

Table 2 needs more explanation. I suggest that explain the difference between regimens and compare the literature progression time with the trial data.

The Dose-response relationship between LDH and risk of CRPC occurrence section needs more detail about the distribution of LDH and authors may want to consider correlate the log scale LDH with CRPC.

The Survival analysis of CRPC occurrence in patients with different LDH levels section need more justification. I suggest that you add more justification on the LDH cut-off (where I believe this is from the restricted spline, but it is not clearly stated). Also, explain the reason why not treat LDH as continuous variable.

Additional comments

Regarding the data analysis method, the author could think of propensity score weighting to estimate the treatment effect of different regimens.

---

## Round 0.2 · accepted · Accept

Thanks for the revision and answer to the reviewers. We have received two positive reviews now. There are no remaining critical remarks.

Reviewer 1 ·

Basic reporting

The paper is written in easy-to-read English and the data is also without issue.

Experimental design

This paper is very clear in its methodology and objectives. The issues previously pointed out have also been well addressed.

Validity of the findings

The characteristics of the paper are also well articulated.

Reviewer 3 ·

Basic reporting

The author add more information in the introduction.

Experimental design

The author updates the wording in design.

Validity of the findings

No further comments.